# A Non-Contact Detection Method for Multi-Person Vital Signs Based on IR-UWB Radar

**DOI:** 10.3390/s22166116

**Published:** 2022-08-16

**Authors:** Xiaochao Dang, Jinlong Zhang, Zhanjun Hao

**Affiliations:** 1College of Computer Science & Engineering, Northwest Normal University, Lanzhou 730071, China; 2Gansu Province Internet of Things Engineering Research Center, Lanzhou 730070, China

**Keywords:** IR-UWB radar non-contact, vital signs, smoothing splines, CIR, crossing threshold

## Abstract

With the vigorous development of ubiquitous sensing technology, an increasing number of scholars pay attention to non-contact vital signs (e.g., Respiration Rate (RR) and Heart Rate (HR)) detection for physical health. Since Impulse Radio Ultra-Wide Band (IR-UWB) technology has good characteristics, such as non-invasive, high penetration, accurate ranging, low power, and low cost, it makes the technology more suitable for non-contact vital signs detection. Therefore, a non-contact multi-human vital signs detection method based on IR-UWB radar is proposed in this paper. By using this technique, the realm of multi-target detection is opened up to even more targets for subjects than the more conventional single target. We used an optimized algorithm CIR-SS based on the channel impulse response (CIR) smoothing spline method to solve the problem that existing algorithms cannot effectively separate and extract respiratory and heartbeat signals. Also in our study, the effectiveness of the algorithm was analyzed using the Bland–Altman consistency analysis statistical method with the algorithm’s respiratory and heart rate estimation errors of 5.14% and 4.87%, respectively, indicating a high accuracy and precision. The experimental results showed that our proposed method provides a highly accurate, easy-to-implement, and highly robust solution in the field of non-contact multi-person vital signs detection.

## 1. Introduction

Since December 2019, the epidemic and outbreak of Corona Virus Disease 2019 (COVID-19) has become the most serious global health problem and has had a huge impact on health care systems worldwide. COVID-19 has strong propagation characteristics [1]. When volunteers and medical workers check patients’ vital signs, they often make contact with patients or secretions, resulting in personal protective equipment (Personal Protective Equipment, PPE) surface residual contact with live viruses left by patients. This situation greatly increases the risk of infection [1]. IR-UWB technology has good characteristics, such as non-invasive, higher penetration, accurate ranging, low power, low cost, simple hardware, and robustness to multipath interference. Therefore, IR-UWB radar technology is more suitable for the field of non-contact vital signs detection.

Ultra-Wide Band (UWB) radar is a vital telemetry technology for life detection and non-contact monitoring for its low power consumption, high penetration capability, and strong anti-interference ability, as well as high resolution. It has been extensively employed in earthquake disaster rescue, home monitoring, military medicine, and many other areas over the past few years. UWB technology has been gradually adopted since the Federal Communications Commission (FCC) approved the civil microwave range of 3.1 G–10.6 GHz in ‘02 [2]. UWB technology has been applied to various aspects of life (e.g., vital signs monitoring, through-wall detection, trajectory tracking [3], and indoor positioning [4]). There are two sets of definitions for UWB signals. One set is defined as absolute bandwidth higher than 500 MHz, while the other set is defined as fractional bandwidth higher than 20% [2].

UWB radar is capable of detecting human targets by sensing the micro-motion information on the body’s surface caused by breathing and heartbeat [5]. Radar transmits electromagnetic waves from the radar transmitter, and the waves reach the human body through the propagation medium. In addition, the electromagnetic pulses are scattered by the human body to generate the corresponding echo signals, which are propagated through the medium to the radar receiver. Next, the signals are received by the receiver and then sampled for data. IR-UWB radar shows the advantages of smaller volume, fine power consumption, high SNR, and high anti-clutter performance in complex environments [6], compared with microwave Doppler radar. IR-UWB radar is capable of detecting macroscopic and microscopic movements of the human body, and the non-contact ability to estimate vital signs takes on a great significance in medical applications. IR-UWB has been applied to many application scenarios for its high penetration performance in harsh environments, robustness, and accuracy at the centimeter level. Since radar signals generate signal attenuation and distortion when penetrating buildings (e.g., walls), it may result in false alarms and low detection rates. In literature [7], multi-target was achieved by detecting the possibility of target presence at a location in the target’s movement for localization and tracking. A non-contact detection has been confirmed as a good option in complex scenarios where wired connections are not available or difficult to use (e.g., monitoring infants and children) and natural disaster rescue (e.g., earthquake mudslides), as well as vital signs monitoring of burn victims [8]. IR-UWB radar signals have the main advantage of is good material penetration, which can easily penetrate walls for vital signs detection and target identification [9]. In the literature [10], UWB bio-radar was studied, the vital signs imaging model was proposed, and the self-focused imaging method was adopted to implement respiratory rate and heart rate monitoring. In [11], Principal Component Analysis (PCA) was conducted to project and compress the principal components of the data acquired from UWB radar to improve the SNR and obtain the vital signs of the subjected target.

Human vital signs can result in micro-movements in the human chest wall, producing weak echo signals. The original echo information collected by IR-UWB radar systems often contains a considerable number of interfering signals (e.g., linear trends and static clutters, and harmonic interference). Numerous relevant studies worldwide have been conducted to suppress the clutters. In the literature [12], the adaptive clutter cancellation technique was adopted to successfully eliminate a large amount of clutter similar to that of breathing. In both [13,14], the fast Fourier transform (FFT) was initially used, and later the Hilbert–Yellow transform algorithm was adopted to investigate the characteristic relationship between time and frequency of respiration. In [15], singular value decomposition was explored to extract respiration under low SNR conditions. In [16], the radar echo signal was extracted using Empirical Mode Decomposition (EMD), followed by independent principal component analysis for clutter suppression. Subsequently, the radar through-wall detection was investigated. In [17], the ensemble empirical mode decomposition method (EEMD) was employed for efficient extraction and separation of signals, and the first valley peak was analyzed as a temporal feature.

Many UWB radar-based algorithms for non-contact vital signs detection have been proposed over the past few years [18]. Variational Mode Decomposition (VMD) and convolutional sparse coding method were adopted to extract RR and HR [19]. In [20,21], the radar echo signal in the slow time domain was converted into a spectrum, and then RR and HR were extracted for the useful features in the extracted spectrum. An algorithm based on integrated EMD and continuous wavelet filtering was adopted to extract relevant information (e.g., physical signs [22]). A novel method was proposed in literature [23] to extract vital signs information, which was investigated using Quadrature Demodulation (QD) technique to determine the phase of the signal from the input and output. Subsequently, the spectrum averaged harmonic path approach (SHAPA) has been adopted to detect car drivers’ respiration rate and heart rate [24]. In [25], a harmonic path (HAPA) algorithm and SHAPA algorithm were proposed to detect the vital signs information. However, all the above algorithms are considered vital signs detection methods for a single subject target. For multi-person vital signs detection, a two-dimensional (2D) image approach was used in [26] to achieve information regarding the location of the person and relevant information (e.g., breathing for multiple targets). In [27], a single effective peak cluster from adjacent peaks was formed, the effective peak cluster caused by multiple people was detected, and clustering algorithms were executed to detect multi-target vital signs effectively. In [28], tracking and vital signs detection of multiple moving targets were achieved, which can eliminate the adverse effects of body movements. In [29], Variational Mode Decomposition (VMD) was used to decompose the vital signs of different targets into different sub-signals, and multi-target tracking and vital signs detection were achieved. In [30], the VMD algorithm was also adopted to detect the vital signs of multiple targets, whereas it is experimentally proven to have good detection effect and robustness only when the number of targets is two. However, the above algorithms for multiple human vitals detection lack accuracy and algorithm complexity, and they cannot accurately estimate the location information of multiple people, the respiratory rate, as well as the heart rate.

In response to the complexity of existing research worldwide on IR-UWB radar signal analysis, it is difficult to extract multi-person vital signs related information accurately, and range estimation from low SNR received signals using existing methods. On that basis, an effective algorithm CIR-SS based on the channel impulse response (CIR) smoothing spline method is proposed in this study, which is capable of accurately detecting the vital signs of targets even under low SNR conditions and complex environments. The radar echo signals are also subjected to crossing threshold and multi-person TOA distance estimation and CIR-based azimuth to obtain the position information of multiple targets individually. Subsequently, their vital signs are detected separately for multi-person vital signs detection.

The rest of this study is organized as follows. In Section 2, the IR-UWB radar signal is modeled. In Section 3, the proposed algorithm is presented to estimate RR and HR for vital signs detection in this study. In Section 4, the experimental results and performance analysis are presented. Conclusions are presented in Section 5.

## 2. Related Work

### 2.1. IR-UWB Radar Signal Model

The transmitting antenna emits electromagnetic pulses, thus leading to the formation of reflected pulses upon contact with the human body. The micro-motion information of the thorax can be evaluated by the amplitude variation of the reflected pulse and the TOA estimation. On that basis, the respiratory frequency and amplitude information can be analyzed, and the target range estimation can be conducted by algorithms. Thus, the distance from the transceiver antenna to the detected target is derived from the literature [6] and can be expressed as:(1)$$d\left(t\right)=d_{0}+r\left(t\right)=d_{0}+S_{r}\left(t\right)+S_{h}\left(t\right)=d_{0}+A_{r}\sin\left(2\pi f_{r}t\right)+A_{h}\sin\left(2\pi f_{h}t\right)$$
where $d_{0}$ denotes the nominal distance from the transceiver antenna to the human chest; $S_{r}\left(t\right)$ and $S_{h}\left(t\right)$ represent the distance variation due to breathing and heartbeat, respectively; $A_{r}$ and $A_{h}$ represent the displacement amplitude and heartbeat frequency amplitude due to breathing, respectively. $f_{r}$ and $f_{h}$ are the breathing frequency and heartbeat frequency. If $\delta \left(t\right)$ represents the normalized received impulse, the total impulse response is expressed as:(2)$$r\left(\tau ,t\right)=a_{v}\delta \left(\tau -\tau_{v}\left(t\right)\right)+\sum_{i}a_{i}\delta \left(\tau -\tau_{i}\right)$$
where t denotes the observation time; τ denotes the propagation time; and $a_{v}\delta \left(\tau -\tau_{v}\left(t\right)\right)$ denotes the impulse response with propagation time $\tau_{v}\left(t\right)$ and amplitude $a_{v}$ formed by the small movements of the chest wall. $\sum_{i}a_{i}\delta \left(\tau -\tau_{i}\right)$ denotes the corresponding sum of all static targets at propagation time $\tau_{i}$ with amplitude $a_{i}$ by i static targets. The propagation time $\tau_{v}\left(t\right)$ is obtained by $d\left(t\right)$ in (1), as expressed in Equation (3).
(3)$$\tau_{v}\left(t\right)=\frac{2d\left(t\right)}{C}=\tau_{0}+\tau_{r}\sin\left(2\pi f_{r}t\right)+\tau_{h}\sin\left(2\pi f_{h}t\right)$$
where $\tau_{0}=2d_{0}/C$, $\tau_{r}=2A_{r}/C$, $\tau_{h}=2A_{h}/C$, and C is the speed of light, i.e., $3.0\times 10^{8}m/s$. The received signal is
(4)$$R\left(\tau ,t\right)=s\left(\tau \right)\times h\left(t,\tau \right)=a_{v}s\left(\tau -\tau_{v}\left(t\right)\right)+\sum_{i}a_{i}s\left(\tau -\tau_{i}\right)$$

$s\left(\tau \right)$ and × propagate the signal and convolution operation, respectively.

### 2.2. Echo Model

The IR-UWB radar signal sampling process requires sampling of the time domain information, which usually involves two-time axes. One is the fast time domain, i.e., the time axis of the time domain information propagated by a single pulse signal. The other is the slow time domain, which is the time axis adopted to represent the sequential relationship between pulses. To simplify the model assuming the ideal case (i.e., ignoring static echoes and other clutter), Equation (2) is transformed into a discrete-time 2-dimensional echo matrix, which can be expressed as:(5)$$R\left[m,n\right]=r\left(t=mT_{s},\tau =nT_{f}\right)$$
where m and n denote the number of samples in the slow and fast time domains, respectively. $T_{s}$ and $T_{f}$ represent the sampling intervals in the slow and fast time domains.

Equation (4) can be expressed after discretization as
(6)$$\begin{array}{ll} R\left[m,n\right] & =r\left(m\mu_{T},nT_{S}\right)\\ & =a_{v}s\left(m\mu_{T}-\tau_{v}\left(nT_{S}\right)\right)+\sum_{i}a_{i}S\left(m\mu_{T}-\tau_{i}\right)\\ & =a_{v}s\left(m\mu_{R}-v\tau_{v}\left(nT_{S}\right)\right)+\sum_{i}a_{i}S\left(m\mu_{T}-v\tau_{i}/2\right)\\ & =h\left[m,n\right]+c\left[m\right] \end{array}$$
where $\mu_{T}$ denotes the fast-sampling interval; and $T_{S}$ represents the continuous pulse time. m and n express the number of fast time samples and the number of slow time-domain samples, respectively. $v\mu_{T}=2\mu_{R}$, $h\left[m,n\right]$ denotes the human body micro-motion information. $c\left[m\right]$ is other slow time invariant static clutter. Moreover, in the actual experimental process, the received signal may contain such as linear trend, AWGN, non-stationary clutter and other unknown clutter. Thus, the received signal is expressed as:(7)$$R\left[m,n\right]=h\left[m,n\right]+c\left[m\right]+l\left[m,n\right]+w\left[m,n\right]+r\left[m,n\right]+u\left[m,n\right]$$

$l\left[m,n\right]$ is linear trend; $w\left[m,n\right]$ denotes AWGN; $r\left[m,n\right]$ is non-stationary clutter; and $u\left[m,n\right]$ denotes other unknown clutter. Since the radar echo contains a host of interfering signals, how to solve the above interfering factors is elucidated in the algorithm design in Part 3.

The ideal radar echo signal in the static environment after the removal of all clutters is expressed as Equation (4).
(8)$$ℜ\left(\tau ,t\right)=a_{v}s\left(\tau -\tau_{v}\left(t\right)\right)$$

To obtain the respiratory frequency $f_{r}$ and the heart rate $f_{h}$. In slow-time domain, the Fourier transform (FT) of $ℜ\left(\tau ,t\right)$ is
(9)$$Y_{m}\left(f\right)=\int_{-\infty }^{+\infty }ℜ\left(m\delta_{T},t\right)e^{-j2\pi ft}dt$$

Equation (9) is expressed in 2D FT as:(10)$$ϒ\left(m\delta_{T},f\right)=\int_{-\infty }^{+\infty }ϒ\left(v,f\right)e^{-j2\pi v\tau }dv$$
where $ϒ\left(v,f\right)$:(11)$$\begin{array}{ll} ϒ\left(v,f\right) & =\int_{-\infty }^{+\infty }\int_{-\infty }^{+\infty }ℜ\left(m\delta_{T},t\right)e^{-j2\pi ft}e^{-j2\pi v\tau }dtd\tau \\ & =\int_{-\infty }^{+\infty }a_{v}U\left(v\right)e^{-j2\pi ft}e^{-j2\pi v\tau_{v}\left(t\right)}dt\\ & =a_{v}U\left(v\right)e^{-j2\pi v\tau_{0}}\int_{-\infty }^{+\infty }e^{-j2\pi vm_{b}\sin\left(2\pi f_{r}t\right)}e^{-j2\pi vm_{h}\sin\left(2\pi f_{h}t\right)}e^{-j2\pi ft}dt \end{array}$$
where $U\left(v\right)$ denotes the IR-UWB pulses in the fast time domain after FT. f and v denote the spectra in the slow and fast time domains, respectively. Using the Bessel function, Equation (11) can be expressed as
(12)$$ϒ\left(v,f\right)=a_{v}U\left(v\right)e^{-j2\pi v\tau_{0}}\int_{-\infty }^{+\infty }\left(\sum_{k=-\infty }^{+\infty }J_{k}\left(\beta_{r}v\right)e^{-j2\pi kf_{r}t}\right)\left(\sum_{l=-\infty }^{+\infty }J_{l}\left(\beta_{h}v\right)e^{-j2\pi kf_{h}t}\right)e^{-j2\pi ft}dt$$
(13)$$e^{-jz\sin\left(2\pi f_{0}t\right)}=\sum_{k=-\infty }^{+\infty }J_{k}\left(z\right)e^{-j2\pi kf_{0}t}$$

$\beta_{r}=2\pi A_{r}$ and $\beta_{h}=2\pi A_{h}$, so Equation (10) can be expressed as
(14)$$ϒ\left(m\delta_{T},f\right)=a_{v}\sum_{k=-\infty }^{+\infty }\sum_{l=-\infty }^{+\infty }G_{K}\left(\tau \right)\delta \left(f-kf_{r}-lf_{h}\right)$$
where
(15)$$G_{\mathrm{Kl}}\left(\tau \right)=\int_{-\infty }^{+\infty }U\left(v\right)J_{k}\left(\beta_{r}v\right)J_{l}\left(\beta_{h}v\right)e^{j2\pi v\left(\tau -\tau_{0}\right)}dv$$
when $m\delta_{T}=\tau_{0}$, Equation (15) can be yielded as the maximum value, expressed as
(16)$$C_{k}=G_{k}\left(\tau_{0}\right)=\int_{-\infty }^{+\infty }U\left(v\right)J_{k}\left(\beta_{r}v\right)J_{l}\left(\beta_{h}v\right)dv$$
(17)$$ϒ\left(\tau_{0},f\right)=a_{v}\sum_{k=-\infty }^{+\infty }\sum_{l=-\infty }^{+\infty }C_{kl}\delta \left(f-kf_{r}-lf_{h}\right)$$

Set k and l in Equations (16) and (17) as 0, respectively, so, $f_{r}$ and $f_{h}$ can be obtained.

## 3. Proposed Method

The original IR-UWB radar echo signal is first subjected to linear trend cancellation and subsequently filtered by two fifth-order Butterworth filters. A smoothing filter is adopted to obtain a smooth sampled echo signal. Cross-threshold-based multi-person TOA distance estimation and CIR-based azimuth are capable of obtaining the human position information extracted from the radar sampling echo signal. FT is performed in the slow time domain followed by window selection for respiration frequency estimation of individual subject targets. Moreover, the CIR-SS is adopted to eliminate the harmonic signal, thus obtaining the heartbeat frequency of a single subject target in the pure heartbeat signal. Lastly, the vital signs information of multiple people is output by comparing the target information. Figure 1 illustrates the system flow chart.

### 3.1. Clutter Suppression

#### 3.1.1. Static Clutter Suppression

IR-UWB echo signal contains micro-motion information of the test target and the signal received by the surrounding environment through reflection and scattering. A significant influencing factor is static clutter, which can be expressed as:(18)$$J_{c}=\frac{\sum_{m=1}^{M}\sum_{n=1}^{N}R\left[m,n\right]}{M\times N}$$

After its elimination as
(19)$$\Omega \left[m,n\right]=R\left[m,n\right]-J_{c}$$

The linear trend in the echoes can be effectively eliminated using the LTS algorithm [14].
(20)$$W=\Omega^{T}-y{\left(y^{T}y\right)}^{-1}y^{T}\Omega^{T}$$
where $Y=\left[y_{1},y_{2}\right]$, $y_{1}={\left[0,1,\ldots ,N-1\right]}^{T}$, $y_{2}={\left[1,1,\ldots ,1\right]}_{N}^{T}$, T is the transpose matrix.

#### 3.1.2. Clutter Signal Suppression

The experimental environment determines the IR-UWB radar pulse-echo signal, the azimuth angle between the detection target and the antenna, the dielectric constant, the humidity, and the electromagnetic wave polarization. Hence, its value is difficult to determine as suggested in literature [18]. Since we cannot predict the above parameters accurately, bandpass filtering is used instead of matched filtering. In this study, a Butterworth filter is used for filtering with a transfer function:(21)$${\left|H\left(\omega \right)\right|}^{2}=\frac{1}{1+{\left(\omega /\omega_{\varepsilon }\right)}^{2N_{f}}}$$
where $\omega_{\varepsilon }$ and $N_{f}$ denote the cutoff frequency and the order of the filter, respectively. It is experimentally proven that the higher the filter order, the better the performance of the filter will be. Thus, for complexity as low as possible, this study uses two fifth-order Butterworth filters, i.e., $N_{f}=5$, while setting the normalized cutoff frequencies of 0.1036 and 0.0212 for the low-pass and high-pass filters, respectively. The normalized cutoff frequency can be expressed as:(22)$$\omega_{n\varepsilon }=\frac{\omega_{\varepsilon }}{f_{s}}$$

$f_{s}$ denotes the sampling frequency in the fast time domain, and for each index n in the slow time domain after filtering on $W_{m\times n}$ in the fast time domain, there are
(23)$$\begin{array}{l} T\left[m,n\right]=b_{1}W\left[m,n\right]+b_{2}W\left[m-1,n\right]+\ldots +b_{N_{b}+1}W\left[m-N_{b},n\right]\\ -a_{2}W\left[m-1,n\right]-\ldots -a_{N_{a}+1}W\left[m-N_{a},n\right] \end{array}$$

In this study, a 5th order Butterworth filter is adopted, so $N_{b}=N_{a}=5$ is set. Where $a_{i}$ and $b_{i}$ denote the filter coefficients. A smoothing filter for filtering is adopted to suppress non-stationary clutter, i.e.,
(24)$$S\left[k,n\right]=\frac{1}{\lambda }\sum_{m=\lambda k}^{\lambda \left(k+1\right)-1}T\left[m,n\right]$$
where $k=\left[1,\left\lfloor M/\lambda \right\rfloor \right]$, $\left\lfloor M/\lambda \right\rfloor$ denotes the largest integer less then M/λ. To improve the SNR of the signal, the smoothing filter is employed to take an average of 7 values in the slow time domain, so λ=7.

### 3.2. Target Range Estimation

After the previous filtering, most of the clutter is suppressed, whereas in Equation (7), Gaussian noise $w\left[m,n\right]$ is the main factor for the radar echo signal. In this section, we use the statistical characteristic of the received signal for range estimation. Spectral kurtosis has been used over the past few years to extract non-Gaussian signals, while their positions in the frequency domain can be determined. This study uses an optimized algorithm for estimating target spectrum range with Root Mean square (RMS) and Excess Kurtosis (EK). The EK for each fast time domain is given as.
(25)$$E_{kurt}=\frac{\kappa_{4}}{\kappa_{2}^{2}}-3=\frac{E\left[{\left(S\left[m,n\right]\right)}^{4}\right]}{{\left\{E\left[S{\left[m,n\right]}^{2}\right]\right\}}^{2}}-3$$
where $\kappa_{4}$ and $\kappa_{2}$ denote the fourth-order centroid and second-order centroid of the sample, respectively; E denotes the expectation of the sample. The RMS of $S\left[m,n\right]$ is expressed as:(26)$$RMS=\sqrt{\frac{\sum_{n=1}^{N}S{\left[m,n\right]}^{2}}{N}}$$

The RMS of EK is defined as
(27)$$\varphi =E_{kurt}/RMS$$

This statistical feature can only represent the spectral features caused by the target and cannot directly estimate the target range, so further signal processing is required. To further obtain the position information of the tested target from the spectral range, the traditional method uses the Short Time Fourier Transform (STFT) algorithm [31]. In the article [32], STFT and Hamming window are used for vital signs detection. However, STFT is primarily dependent on the time width of the signal for analysis, the time width is difficult to determine in practical experiments. Wavelet transform (WT) does not have the above problems, and it exhibits a variable window size more suitable for analyzing some non-stationary signals. Given a time–domain signal $T\left(\tau \right)$, the continuous WT can be expressed as:(28)$$C_{\tau }=a^{-\frac{1}{2}}\int_{-\infty }^{+\infty }T\left(\tau \right)\bar{\psi }\left(\frac{\tau -b}{a}\right)d\tau$$

Equation $\psi \left(\left(\tau -b\right)/a\right)$ denotes a wavelet with scaling and translation parameters a and b, respectively, and $\bar{\psi }\left(t\right)$ denotes the conjugate complex of the wavelet’s mother function. In this study, the mother wavelet function is chosen to the Mexican Hat (MH) wavelet to obtain spectral information more intuitively, that is
(29)$$\bar{\psi }\left(\tau \right)=\frac{2}{\sqrt{3}}\pi^{\frac{1}{4}}\left(1-\tau^{2}\right)e^{-\frac{\tau^{2}}{2}}$$

Its Discrete Wavelet Transformation (DWT) is
(30)$$D_{\tau }=a^{-\frac{1}{2}}\sum_{n}T\left(\tau \right)\bar{\psi }\left(\frac{\tau -b}{a}\right)$$

Setting the frequency window width from 0.6 GHz to 1.2 GHz to obtain the range estimate of the target object from the echo signal, and then the distance between the target and the IR-UWB radar is estimated as
(31)$$\hat{R}=\frac{v\hat{\tau }}{2}$$
where $\hat{\tau }$ represents the maximum TOA estimate for the corresponding matrix. Thus, the target information can be estimated accurately by the algorithm.

### 3.3. Signal Separation Algorithm CIR-SS

After clutter suppression, the original radar pulse-echo signal is extracted, whereas the thoracic micro-motion caused by the heartbeat is fainter than the respiration, and the heartbeat signal and the higher harmonics of the respiration have the same frequency. The frequency of the heartbeat signal is consistent with the frequency of the higher harmonics of respiration (3rd and 4th). Thus, it is necessary to propose an algorithm to separate respiration and heartbeat effectively. In this study, an optimized algorithm using Smoothing Spline for the CIR of respiration signals is proposed to extract respiration and heartbeat signals effectively from IR-UWB radar echo signals.

A set of initial values of the window is set as follows:(32)$$\left\{t_{i},z\left(t_{i}\right):i=t_{0},t_{0}+T_{s},\cdots ,t_{0}+W\right\}$$

The time between samples is denoted as $T_{s}=1/F_{s}$. W represents the window size, and $t_{0}$ is the initial time. Since the respiration signal has a larger amplitude and resolution. Therefore, the echo signal $S_{r}\left(t\right)$ is written as:(33)$$S_{r}\left(t\right)=\underset{\overset{\wedge }{f}}{\min}\sum_{t=t_{0}}^{t_{0}+W}{\left\{z\left(t_{i}\right)-\overset{\wedge }{f}\left(t_{i}\right)\right\}}^{2}+\lambda \int \overset{\wedge }{f}\prime \prime {\left(t\right)}^{2}dt$$
where λ represents a non-negative smoothing parameter. $\overset{\wedge }{f}$ denotes the estimate of $S_{r}\left(t\right)$, which is expressed as:(34)$$\overset{\wedge }{f}\left(t\right)=\sum_{t=t_{0}}^{t_{0}+W}\overset{\wedge }{f}\prime \;\left(t_{i}\right)f_{i}\left(t\right)$$
where $f_{i}\left(t\right)$ in Equation (34) is a set of spline basis function. The B-Spline Basis Function is adopted as the spline basis function. To obtain the solution of Equation (33), the vector $\overset{\wedge }{\Omega }={\left[\overset{\wedge }{f}\left(t_{0}\right),\cdots ,\overset{\wedge }{f}\left(t_{0}+W\right)\right]}^{T}$ is defined first. Subsequently, the roughness penalty of the spline bases is
(35)$$\int {\overset{\wedge }{f}}^{\prime \prime }{\left(t\right)}^{2}dt={\overset{\wedge }{\Omega }}^{T}A\overset{\wedge }{\Omega }$$
where A is $\int f_{i}{}^{\prime \prime }\left(t\right)f_{j}{}^{\prime \prime }\left(t\right)dt$. In this study, the roughness penalty smooth model with appropriate transformation of the B-sample basis function is applied to the smooth processing of data with constraints to achieve a smooth fitting effect in radar echo data. Thus, the penalized sum-of-squares can be written as:(36)$$\underset{\overset{\wedge }{\Omega }}{\min}{\left\{z-\overset{\wedge }{\Omega }\right\}}^{T}\left\{z-\overset{\wedge }{\Omega }\right\}+\lambda {\overset{\wedge }{\Omega }}^{T}A\overset{\wedge }{\Omega }$$
where $y={\left\{z\left(t_{0}\right),\cdots ,z\left(t_{0}+W\right)\right\}}^{T}$, and the minimum value of Equation (36) is $\overset{\wedge }{\Omega }={\left(I+\lambda A\right)}^{-1}z$, so the heartbeat signal with the elimination of the breathing signal is expressed as:(37)$$\overset{\wedge }{S_{h}}\left(t\right)=y\left(t\right)-{\overset{\wedge }{m}}^{T}\times f\left(t\right)$$

### 3.4. Multi-Person TOA Estimation Algorithm Based on Threshold Crossing

Since the sampled signal contains a host of dense information, the conventional methods cannot determine the location information, a multi-person TOA estimation algorithm based on threshold crossing is proposed in this study. The algorithm can find the effective peak of the received signal by setting the threshold in the received signal to determine the respective person’s location information and match the vital signs information of each person by comparing the relevant information.

The IR-UWB radar signal is divided into several coherent clusters, so there is one and only one local maximum peak in the respective coherent cluster. The maximum local peak in each coherent cluster is considered a valid peak and subsequently compare each local peak with its neighboring local peaks through a series of recursive algorithms to determine the coherent clusters. Lastly, the number of coherent clusters we obtain refers to the number of detected persons, and the position of each valid peak is the position of a single person. A simple TOA is capable of estimating the relevant location information. The multi-person TOA based on the threshold crossing estimation algorithm is specified as follows.Set horizontal thresholds $T_{hd}$, left time $N_{left}$ and right time $N_{right}$;Compare the instantaneous power $p\left(t\right)$, $p\left(t\right)=r^{2}\left(t\right)$, where $r\left(t\right)$ is the original signal;Initialize. Set the dirty map $d_{0}\left(t\right)$ by the power signal; set the sequence $A_{toa}\left(1:end\right)=0$ of TOA, where $\left(a:b\right)$ means from a to b; also initialize the number of people k=0 and the number of iterations n=0;Find the index ${\hat{\tau }}_{n}=\arg\max_{t}\left(d_{n}\left(t\right)\right)$, whose magnitude is ${\hat{a}}^{’}{}_{n}=d_{n}\left({\hat{\tau }}_{n}\right)$;Determine if ${\hat{a}}^{’}{}_{n}<T_{hd}$, then go to step 9;Update the dirty map $d_{n+1}\left(t\right)=d_{n}\left(t\right)$ by filling it with 0: $d_{n}\left({\hat{\tau }}_{n}-N_{left}:{\hat{\tau }}_{n}+N_{right}\right)=0$;If ${\hat{a}}^{’}{}_{n}<\max\left\{r\left({\hat{\tau }}_{n}-N_{left}:{\hat{\tau }}_{n}+N_{right}\right)\right\}$, we update its iteration number n=n+1 while skipping to step 4;Storing the TOA information ${\hat{\tau }}_{n}$, $A_{toa}\left(k\right)={\hat{\tau }}_{n}$, while updating the number of iterations and the number of people, n=n+1, k=k+1, and go to step 4;The TOA position information of the respective person is $A_{toa}\left(1:k\right)$.

On that basis, the distance of multiple people can be obtained, i.e., the TOA information of each person. However, we cannot determine the azimuthal position of the distance, so we will model each person to obtain their specific position information in the next subsection.

### 3.5. CIR Based Multi-Person Azimuth Estimation Algorithm

In this part, a correlation model is built to get the azimuth information of a person, and then compare it with the TOA information to determine the specific location information of the target, and then acquire the specific vital signs information (including breathing and heartbeat information) of each person by matching the information. In [33], the authors propose a fully integrated 2-channel beamformer for 3–5 GHz pulsed ultra-wideband (IR-UWB) receivers. The literature [34,35] proposed a UWB-based 4-channel beamformer. In [36], the authors propose a multiple-input multiple-output (MIMO) array that employs multiple IR-UWB radar transceivers. In multi-user scenarios, beamforming technology can suppress real-world interference while increasing resolution, enabling higher data rates, stronger anti-jamming capabilities, and range extension.

The IR-UWB radar used in this paper is a multiple-input multiple-output (MIMO) radar that supports beamforming technology, and the transmitted signal is a multi-carrier Gaussian pulse train signal. The receiver has a low noise amplifier (LNA), three buffer stages, an energy detector, and an 8-bit SAR ADC. It will collect and quantize the received UWB signal energy. The IR-UWB radar system can generate 3D beamforming patterns where both the transmitter and receivers are configured as integrated 2 × 2 2D planar array antennas. Prior knowledge suggests that the CIR of the target is different in different directions, so a signal model can be established using beamforming (BF) technology for its transmitters $T_{x}$ and receivers $R_{x}$ to obtain the target azimuth and other relevant information. The operation is presented as follows. Conventional Beamforming (CBF) technique is used at the transmitter to generate a beam with the orientation angle $\left(\theta ,\phi \right)$, where the coefficient $s_{Tx}$ of the direction vector of the m antenna is expressed as:(38)$$s_{Tx,m}\left(\theta ,\phi \right)=\exp\left[\left(-j2\pi \left(d_{mx}\cos\theta \sin\phi +d_{my}\cos\theta \cos\phi \right)\right)/\lambda_{c}\right]$$
where $d_{my}$ and $d_{mz}$ represent the horizontal and vertical distances, respectively. The CBF technique is also employed at the receiver side corresponding to the transmitter side to receive the signal while obtaining the Angle of Arrival (AOA) of the signal, and the direction vector $s_{Rx}$ of the nth antenna is
(39)$$s_{Rx,n}\left(\theta ,\phi \right)=\exp\left[\left(-j2\pi \left(d_{nx}\cos\theta \sin\phi +d_{ny}\cos\theta \cos\phi \right)\right)/\lambda_{c}\right]$$
where $d_{n,y}$ and $d_{n,z}$ denote the horizontal and vertical distances, respectively. In this study, the BF technique is performed on $T_{x}$ and $R_{x}$.

## 4. Experimental Results and Analysis

### 4.1. Experimental System

The experimental system in this study comprised Novelda-developed X4m200 IR-UWB MIMO radar and a laptop computer. The IR-UWB radar (Figure 2) comprised transmitting antennas $T_{x}$ and receiving antennas $R_{x}$, a radar module, a Microcontroller Unit (MCU), a storage unit, as well as a power supply unit.

#### 4.1.1. Parameters of the Radar

The IR-UWB radar transmitter had a bandwidth of 1.4 GHz, and the sensor center frequency was 7.29 GHz in compliance with FCC regulations. The receiver accepted the return signal at 23.328 GS/s, and it is capable of continuously covering a range of 10 m. The radar device architecture is illustrated in Figure 2b, comprising a memory, processor, pulse generator, LNA, digital baseband, transceiver, analog front-end, power management unit (PMU), and serial peripheral interface (SPI). The pulse generator generated high-frequency pulses and sent them through the transmitting antenna under the control of the processor, and the echoes were generated when they reached the human chest, and the receiving antenna received the radar echoes. The time interval between pulses transmitted by IR-UWB radar was obtained using pulse repetition frequency (PRF).

The data acquisition refers to a laptop computer with Intel i5-9500 CPU and 16 G memory. The computer is connected to the radar control module via a USB interface to perform data acquisition and send control commands to the radar MCU. To compare the experimental results in each case, contact sensors (e.g., Respiration belt and ECG sensor module) served as the benchmark devices to check the vital signs of the subjects. The contact sensor used to obtain respiratory rate was the respiration belt (Gdx-rb, Vernier Software & Technology, 13979 SW Millikan Way, Beaverton, OR, USA), measuring the tension of chest vibrations generated during each breath to obtain respiratory rate and other relevant data. Furthermore, the ECG sensor (Psl-iECG2, PhysioLab Co., Ltd., Hyoyeol-ro 111 Buk-gu Busan 46508 Korea, Busan, South Korea) was adopted to detect the heart rate.

#### 4.1.2. Experimental Scene

Eight subjects were selected for relevant experiments in complex indoor, open indoor, and open outdoor environments, respectively. The proposed CIR-SS algorithm was used for the frequency estimation of signals and the estimation of the subject’s target range.

The complex indoor environment was selected from a laboratory with more noise interference, which had desks, benches, computers, and other furniture, and its size was 5 m × 5 m. The open indoor environment was selected from a relatively empty hall on the first floor of a teaching building, and its size was 5 m × 5 m; the open outdoor environment was selected from an unoccupied outdoor garden, and its size was 5 m × 5 m. The experimental scenario is illustrated in Figure 3 The IR-UWB radar used to get data was placed on a tripod of 80 cm height. The subject was kept relatively still during the experiment, sitting on a bench 40 cm high, with the chest 80 cm above the ground, and the upper body kept straight with the chest facing the IR-UWB radar for even breathing.

When acquiring IR-UWB radar data, different experimental environments, measurement distances, and experimental personnel can affect the echo signal. The basic information of eight subjects (four males and four females) is listed in Table 1.

IR-UWB radar data acquisition set the PRF to 600 KHz, set the average number of acquisitions $N_{a}=30$, through six segments sampled simultaneously, the time window of each segment was set to 124 ns, and the sampling number of each segment N=682. Subsequently, it is easy to know the number of samples in the fast time domain is N∗6=4092. Therefore, each pulse signal reception time was $N*N_{a}/PRF=0.0341\;s$, and 1759 pulses can be received in the 60 s. The data were collected from eight subjects sitting facing the IR-UWB radar, keeping even breathing, at distances of 1 m, 2 m, 3 m, 4 m, and 5 m from the radar, and each group of data was collected five times, respectively, and the valid data duration for each acquisition was 10 min.

To facilitate the performance analysis of the system in the following, the *SNR* of the measured vital signs to noise and clutter are defined as follows.
(40)$$SNR=10\log_{10}\frac{\int_{f_{p-B}}^{f_{p+B}}\left|P_{x}\left(f\right)\right|df}{\int_{V_{R}}^{V_{H}}\left|P_{x}\left(f\right)\right|df-\int_{V_{P-B}}^{V_{P+B}}\left|P_{x}\left(f\right)\right|df}$$
where $V_{R}$ and $V_{H}$ denote the ranges of respiratory and heartbeat frequencies, respectively; $f_{p}$ is the peak index bin; $P_{x}\left(f\right)$ is the spectrum of vital sign, and the peak range is defined as B. To evaluate the error, the following definition is made.
(41)$$Error=\frac{1}{N}\sum \frac{\left|HR_{ref}-HR_{meas}\right|}{\left|HR_{ref}\right|}\times 100\%$$
where $HR_{ref}$ is the true value of the sample measured by pulse counting, $HR_{meas}$ is the estimated value of the sample, and N is the number of samples.

### 4.2. Performance Analysis of Vital Signs Algorithm

#### 4.2.1. Influence of Different Distances

In this study, several sets of comparative experiments were designed to investigate the effect of the subject’s distance from the IR-UWB radar on the radar path loss. Eight subjects were individually tested for vital signs at 1 m, 2 m, 3 m, 4 m, and 5 m from the radar in a complex indoor, open indoor, and open outdoor environment, respectively. In this part, subject A is selected to estimate the respiratory and heartbeat frequencies in different environments and distances, and the experimental results are shown in Figure 4.

As depicted in Figure 4, the subject’s vital signs (including breathing and heart rate) can be accurately estimated using the proposed algorithm when the subject is at the same detection distance from the radar in the same environment. Moreover, it is approximately consistent with the data measured by the sensors. Nevertheless, as the distance of the subject from the radar increased, the error of the estimated data increased significantly. To further compare the effect of different subject distances on the radar echoes, we calculated the average error of respiration and heartbeat for eight subjects in three different environments, and the results are listed in Table 2. As depicted in Table 2, the error of the estimated vital signs increased gradually as the distance of the test target increased. The experimental results showed that the average error measured at 1~3 m was smaller, and the estimated vital signs were closer to the real values. The average errors of respiratory rate were calculated as 4.33%, 4.80%, 4.75%, 5.66%, and 6.72% at 1 m, 2 m, 3 m, 4 m, and 5 m from the radar, respectively; the average errors of heart rate were 5.40%, 5.66%, 6.09%, 6.73%, and 7.35%, respectively. Figure 5 shows the time and frequency domain maps of the vital signs of subject A acquired using IR-UWB radar in an open outdoor environment at 1 m, 2 m, 3 m, 4 m, and 5 m, respectively.

As depicted in Figure 5a, the resolution of the IR-UWB radar acquired vitals signal gradually decreases as the distance increases in the time domain. However, it can still be analyzed in the frequency domain by the proposed algorithm, which shows that the influence of distance on the algorithm in this study is slight, and it also shows that the proposed algorithm shows robustness. As depicted in Table 2, the accuracy rate of this study is high in the distance of 1–3 m, and the average accuracy rates of respiration and heartbeat are calculated as 4.63% and 5.72%, respectively.

#### 4.2.2. Environmental Effects on Vital Signs Signals

In this study, to evaluate the effects of the subject environment on the vital sign signals, experiments were conducted in three different scenarios, including complex indoor, open indoor and open outdoor environments, where eight subjects were individually subjected to vital signs detection at 1 m, 2 m, 3 m, 4 m, and 5 m, respectively, from the IR-UWB radar. Two sets of comparison experiments were designed to demonstrate the effects of the subject environment on the received vital signs. One group was in an indoor environment, and the comparison experiments were performed for subject A in a complex environment in a real room and an open indoor environment, respectively. The data used in the comparison experiments were averaged over 5 min of measurement, measuring vital signs and determining the accuracy of their location estimation, and the results are presented in Figure 6. The other group is the same in the open environment, subject A was subjected to the comparison experiment in the open indoor and open outdoor environment, respectively, and the results are presented in Figure 7.

In the first set of comparative experiments, by observing Figure 6a, it can be seen that when subject A was performing respiratory frequency estimation in an indoor environment, the estimated respiratory frequency in an open environment was closer to the respiratory belt sensor than in a complex environment. The measured data were closer to the real breathing rate of subject A, and the error was smaller. The experiment expects that the multipath effect and other interference signals received by IR-UWB radar in the complex indoor environment were relatively large. Figure 6b is the heartbeat frequency estimation in a complex indoor environment and an open indoor environment. At the same time, compared with the data collected by the ECG sensor, we found that the estimated heartbeat frequency value in an open indoor environment was closer to the value of the ECG sensor. It was closer to the true heartbeat frequency of subject A. Figure 6c is the position estimation of the target with error bars. As depicted in this figure, the proposed algorithm is capable of accurately estimating the position information of the target, and the estimated error is stable at nearly 0.18 after the calculation range. The multipath effect is extremely significant in the complex indoor environment due to debris (e.g., computers, desks, and benches). The vital signs information echo signal obtained by IR-UWB radar contains a considerable number of clutter signals, thus affecting the vital signs estimation and range estimation of the tested target. As depicted in Figure 6c, the range estimation performed in an open indoor environment is more accurate and precise than in a complex indoor environment, and the error is also smaller.

Figure 7 presents the second set of comparative experiments. Subject A was tested for vital signs in both indoor and outdoor environments in the same open environment. As depicted in Figure 7a,b, the subjects’ estimated heart rate and respiratory rate in an outdoor environment are closer to the respiratory belt and ECG sensor values. The possible reason for this result is that in the indoor environment, interference signals (e.g., the reflection of reinforced concrete and walls) affect the IR-UWB radar. As depicted in Figure 7c, using the proposed algorithm in an open outdoor environment can more accurately estimate the position information of the current target to estimate the position of subject A. In other words, it is farther from the baseline in the figure. At the same time, in an open outdoor environment, compared with an open indoor environment, the error of the range estimation is also smaller.

The experimental results show that the proposed algorithm achieves high robustness and accuracy. According to the calculation results, the average respiratory frequency errors in complex indoor, open indoor and open outdoor environments are nearly 6.54%, 5.35%, and 3.87%, respectively; their heartbeat frequency errors are approximately 7.01%, 6.03%, and 5.69%, respectively; and their target range estimation errors are about 0.258, 0.183, and 0.089, respectively.

#### 4.2.3. Impact of Penetrating Medium on the Algorithm

This study considers the effect of the received signal when IR-UWB radar penetrates different media, so the following two sets of comparative experiments were performed. In the first set of experiments, the subjects performed vital signs detection at 1 m, 2 m, 3 m, 4 m, and 5 m from the radar. The relevant vital signs detection experiments were performed with or without walls for analysis. Figure 8 presents the through-wall performance of the proposed algorithm and the experimental results. In another set of experiments, the subjects performed vital signs detection at a position of 2 m. During the experiment, a different medium was placed between the IR-UWB radar and the subject (at 1 m) (e.g., a 10 cm thick wall and a 10 cm thick wooden board). The Line-of-sight (LOS) environment is compared, and the experimental results are illustrated in Figure 9.

In the first set of comparative experiments, by observing Figure 8a, it can be found that as the experimental distance increased, the average detection rate of the detected vital signs decreased in different ranges. When there was no obstacle between the subject and the IR-UWB radar, i.e., under the LOS condition, the accuracy rate was higher than that of the Non-line-of-sight (NLOS) environment. At the same time, the path loss of the radar signal when penetrating the wall was greater than that when the radar signal penetrates the wall. Thus, the accuracy of detecting vital signs in penetrating the wooden board was slightly higher than that of penetrating the wall. Figure 8b presents the cumulative error distribution map, which reveals the cumulative effect of the error. As depicted in the figure, the presence of steel bars and concrete walls in the wall had a greater impact on the IR-UWB radar echo, and the wooden board was second. In the case of LOS, through-board and Through-well, the average accuracy of the subjects’ vital signs was 95.30%, 93.52%, and 92.20%, respectively. Figure 9 presents the time-domain and frequency-domain diagrams of IR-UWB radar signals when they pass through different media. It can be indicated that the vital signs signals have high resolution.

#### 4.2.4. Effect of the Number of Subjects

In this study, the crossing threshold-based multi-person TOA estimation and the CIR directional angle localization algorithm are used for multi-target determination, and then the IR-UWB radar echoes are used to detect vital signs. To further evaluate the effect of the number of subjects in this study on the vital signals received by the IR-UWB radar, the following experiments were conducted: single and multiple human vital signs detection, including breathing and heartbeat, were performed under the same indoor open environment conditions, respectively. Moreover, the experimental scenarios are presented in Figure 10. To further validate the accuracy of the proposed method, a Bland–Altman consistency analysis was conducted to compare ten sets of data values for single, two and three targets, respectively. We performed the Bland–Altman consistency analysis for the respiration data by comparing the estimated beat per minute (bpm) of respiration from the IR-UWB radar with the data obtained from the respiratory belts and the experimental results are presented in Figure 11. For the heartbeat data, we performed a Bland–Altman agreement analysis between the bpm of heartbeat estimated by IR-UWB radar and the ECG data, and the experimental results are shown in Figure 12.

Figure 11 presents the Bland–Altman agreement analysis of the estimated respiratory bpm from IR-UWB radar and respiratory tapes for the different number of subjects, where the horizontal axis represents the mean value of the respiratory bpm estimated from the corresponding IR-UWB radar and the vertical coordinate is the difference between the estimated respiratory bpm and the respiratory bpm from the respiratory tapes. The horizontal axis of the same Figure 12 represents the mean of the heartbeat bpm estimated by the IR-UWB radar and the heartbeat bpm obtained from the ECG, and the vertical coordinate is the difference between the estimated heartbeat and the heartbeat bpm obtained from the ECG. Mean denotes the mean difference between the value estimated from the radar and the value calculated using the data obtained from the contact sensor, and Mean±1.96SD is 1.96 times the standard deviation to make the upper and lower limits. As depicted in Figure 11, the Mean and Mean±1.96SD values increased to varying degrees as the number of people increased, suggesting that the calculated error increases as the number of subjects increases, but all data are within the upper and lower limits of 1.96 times the standard deviation. Figure 12 presents the results of the Bland–Altman agreement analysis between the IR-UWB radar estimated heartbeat data and the contact ECG sensor for the different number of subjects. The experiments showed that multiple targets affected the radar echoes and thus had an effect on the final experimental estimation results. Figure 12c depicts that there was a point where the mean value of 60 exceeded the lower limit of 1.96 times the standard deviation, suggesting that the error is too large due to the weak signal and the interference of other signals when measuring the heartbeat of three targets. Likewise, when the target was a person, the estimated error was the smallest at 0.3074 bpm, and the overall heartbeat error was calculated to be 0.8069 bpm on average. The specific experimental data for error estimation are listed in Table 3.

#### 4.2.5. Performance Analysis of Different Algorithms

To further analyze the algorithm’s performance we proposed, this study compares some typical non-contact vital signs detection methods in recent years and analyzes the error and SNR in the measurement of vital signs. We selected a person with the same respiratory data in the LOS environment, used different algorithms to estimate the respiratory frequency, and drew a time domain diagram (within the 20 s) and a frequency domain diagram. The results are shown in Figure 13. In the same way, different algorithms are used to estimate the heartbeat frequency, and the time domain diagram (within 10 s) and frequency domain diagram are drawn. The result is shown in Figure 14.

Figure 13 is a comparison of breathing estimation using five different algorithms in recent years and this study. As depicted in Figure 13a, the six algorithms on the time domain diagram are approximately the same. In contrast to Figure 13b, the proposed algorithm is more significant on the frequency domain diagram, and the respiratory frequency can be estimated relatively accurately. Likewise, Figure 14 depicts a comparison diagram for heartbeat estimation. In Figure 14a, the proposed algorithm is closer to the real heartbeat, and it also performs well in the frequency domain diagram of Figure 14b. To further compare the performance of different algorithms.

Table 4 lists the error estimation and SNR when different algorithms are used for vital signs. The proposed algorithm has an average error of 5.14% and 4.87% for breathing estimation and heartbeat estimation, respectively. Compared with the other five algorithms, the proposed algorithm reduces the estimated breathing and heartbeat errors by 3.91% and 4.77% on average, while the SNR is increased by nearly 8.75 dB on average. The experimental results suggest that the proposed algorithm achieves high accuracy in detecting breathing and heartbeat frequency. Moreover, this study has a high SNR, suggesting that the proposed algorithm achieves high robustness and can suppress or eliminate the clutter signal in the IR-UWB radar echo signal. The above analyses and findings are the original intentions of this study.

## 5. Conclusions

The purpose of this study is to design algorithms that can effectively eliminate interference signals from IR-UWB radar echo signals, and at the same time, can accurately and effectively estimate the breathing and heartbeat frequencies of the target. The CIR-SS algorithm is proposed to effectively extract and separate the respiration and heartbeat signals due to the low SNR of the received signals when the IR-UWB radar senses human respiration and heartbeat. The multi-person TOA estimation algorithm using Threshold Crossing CIR-based multi-person azimuth estimation algorithm can effectively estimate multiple targets’ location information and compare the target information to achieve multi-person vital signs detection. The algorithm’s performance is analyzed for subjects with different detection ranges and different scenarios, and the average errors of respiratory rate and heart rate estimation reach 5.14% and 4.87%, respectively. Moreover, the proposed algorithm achieves high SNR, high robustness, and easy implementation compared with other vital signs detection methods proposed over the past few years through comparison experiments. Future research will be carried out on the Doppler effect of IR-UWB radar, thereby effectively reducing the algorithm’s complexity. At the same time, due to a restriction set in the experiment in this paper, the subjects are all still, which is not suitable for moving subjects. The future research direction can be vital signs detection for moving subjects.

## Figures and Tables

**Figure 1 sensors-22-06116-f001:**
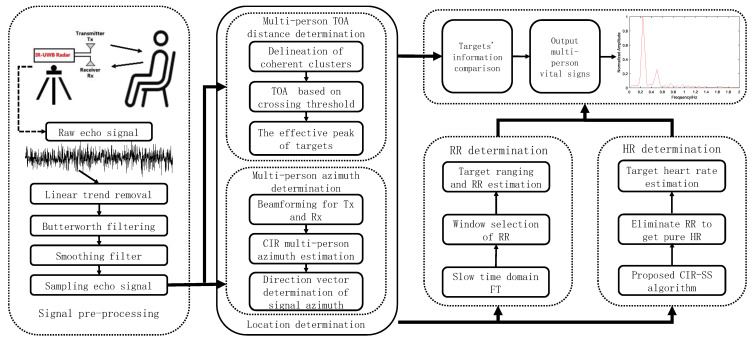
The system flow chart of this study.

**Figure 2 sensors-22-06116-f002:**
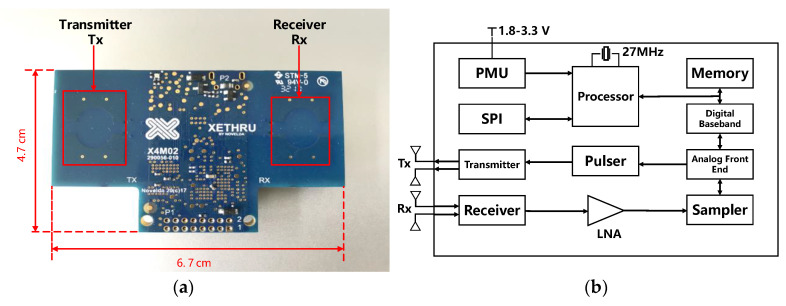
IR-UWB radar equipment and its structure diagram. (**a**) Radar equipment diagram. (**b**) Radar equipment structure diagram.

**Figure 3 sensors-22-06116-f003:**
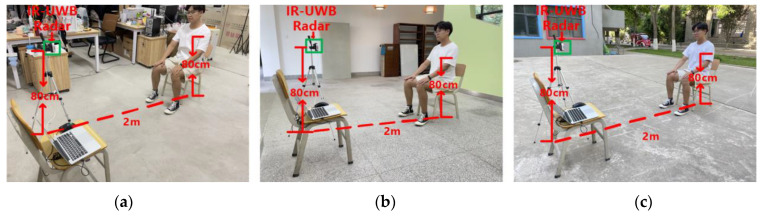
Experimental scene diagram. (**a**) Complex indoor environment. (**b**) Open indoor environment. (**c**) Open indoor environment.

**Figure 4 sensors-22-06116-f004:**
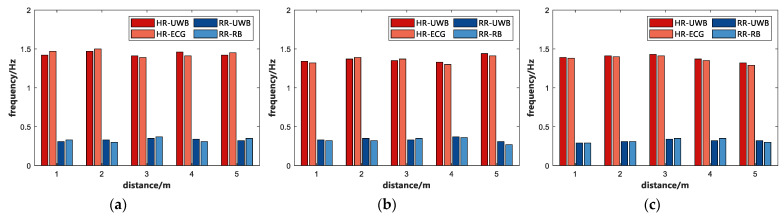
The estimated vital signs of subject A in three different environments and at different distances. (**a**) Complex indoor environment. (**b**) Open indoor environment. (**c**) Open outdoor environment.

**Figure 5 sensors-22-06116-f005:**
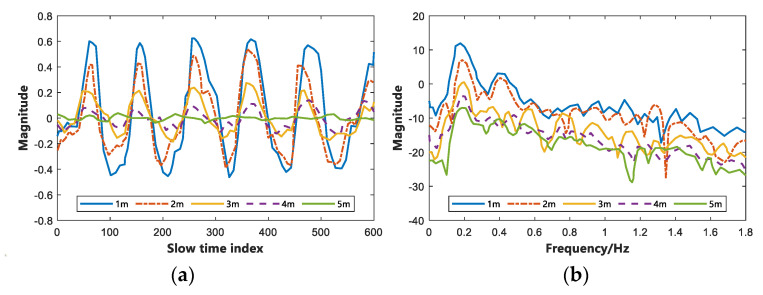
Vital signs signals obtained by IR-UWB radar in Outdoor open environment. (**a**) Time domain at different distances. (**b**) Frequency domain at different distances.

**Figure 6 sensors-22-06116-f006:**
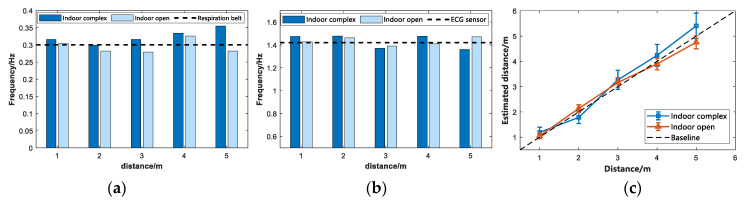
Indoor environment. (**a**) Respiratory rate estimation. (**b**) Heartbeat frequency estimation. (**c**) Range estimation.

**Figure 7 sensors-22-06116-f007:**
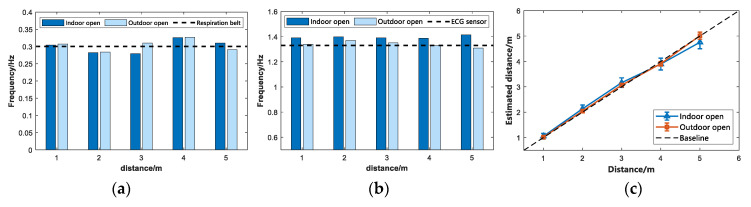
Open environment. (**a**) Respiratory rate estimation. (**b**) Heartbeat frequency estimation. (**c**) Range estimation.

**Figure 8 sensors-22-06116-f008:**
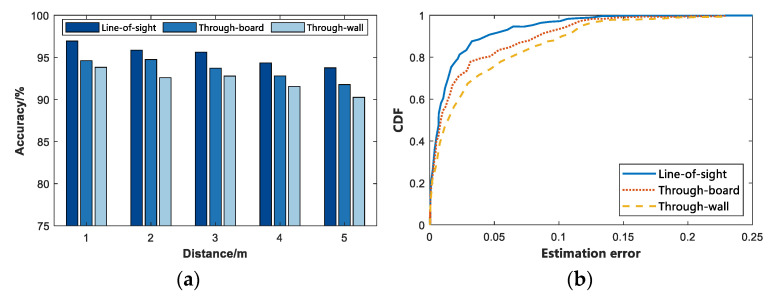
Distribution of accuracy and error accumulation when penetrating different media. (**a**) Accuracy rate. (**b**) Cumulative distribution of errors.

**Figure 9 sensors-22-06116-f009:**
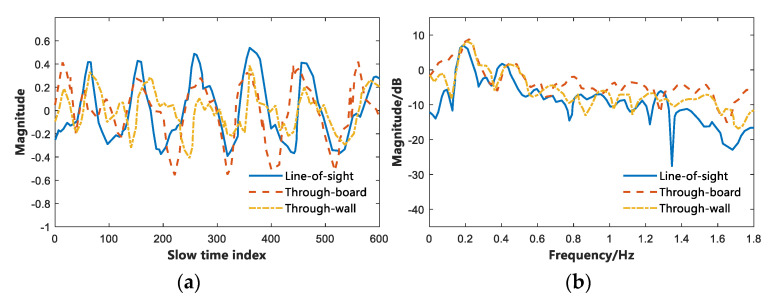
Time domain and frequency domain diagram of penetration through different media. (**a**) Time domain of penetrating different media. (**b**) Frequency domain penetration of different media.

**Figure 10 sensors-22-06116-f010:**
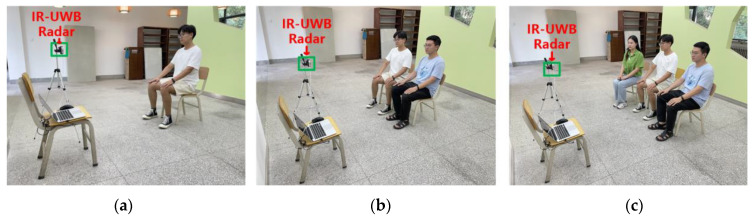
IR-UWB radar experiment scene diagram under different number of tested targets. (**a**) One subject. (**b**) Two subjects. (**c**) Three subjects.

**Figure 11 sensors-22-06116-f011:**
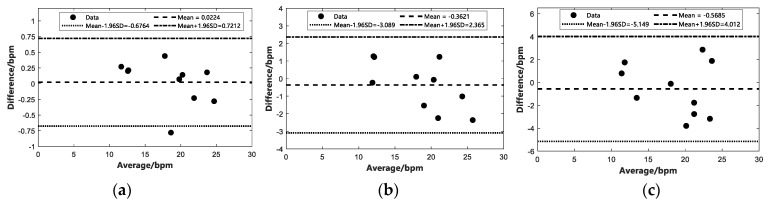
Bland–Altman consistency analysis of respiration measured by IR-UWB radar and respiration belt under different number of targets. (**a**) One subject. (**b**) Two subjects. (**c**) Three subjects.

**Figure 12 sensors-22-06116-f012:**
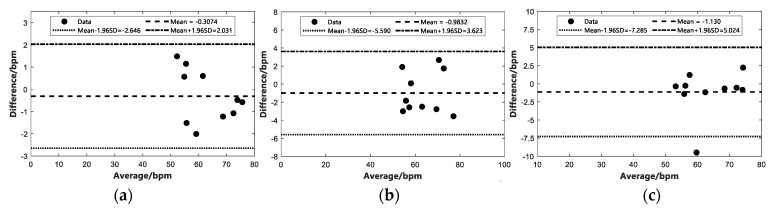
Bland–Altman consistency analysis of heartbeat measured by IR-UWB radar and ECG under different number of targets. (**a**) One subject. (**b**) Two subjects. (**c**) Three subjects.

**Figure 13 sensors-22-06116-f013:**
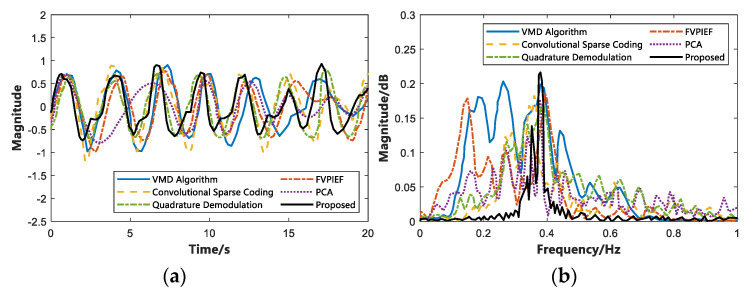
Time domain and frequency domain of breathing estimation using different algorithms. (**a**) Time domain diagram of breathing estimation. (**b**) Frequency domain diagram of breathing estimation.

**Figure 14 sensors-22-06116-f014:**
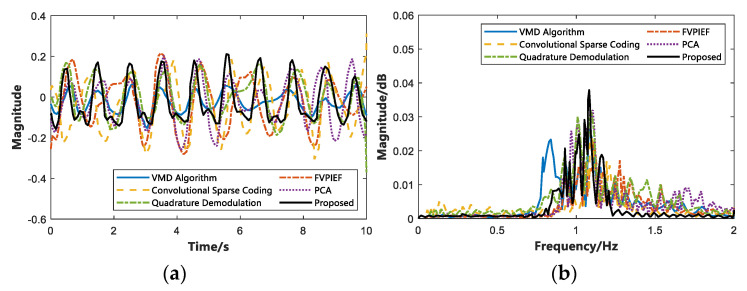
Time domain and frequency domain of heartbeat estimation using different algorithms. (**a**) Time domain diagram of heartbeat estimation. (**b**) Frequency domain diagram of heartbeat estimation.

**Table 1 sensors-22-06116-t001:** Basic information of experimenters.

Subject (Sex)	Age	Height/cm	Weight/kg	Chest Width/cm
A (Male)	24	175	73	103
B (Male)	23	181	86	106
C (Male)	26	165	56	91
D (Male)	32	183	88	98
E (Female)	23	166	63	88
F (Female)	21	163	55	85
G (Female)	24	168	72	96
H (Female)	44	175	70	105

**Table 2 sensors-22-06116-t002:** Vital signs errors and SNR at different distances parameter setting of IR-UWB radar.

			1 m	2 m	3 m	4 m	5 m
Complex indoor	RR	Error	5.24%	6.07%	5.44%	7.21%	8.75%
		SNR(dB)	10.52	9.72	8.54	8.10	7.73
	HR	Error	6.32%	6.48%	6.89%	7.13%	8.20%
		SNR(dB)	5.74	4.31	1.24	−1.33	−2.17
Open indoor	RR	Error	4.75%	4.92%	5.02%	5.35%	6.70%
		SNR(dB)	12.01	11.76	9.67	9.52	8.93
	HR	Error	5.13%	5.37%	5.91%	6.74%	7.02%
		SNR(dB)	7.89	6.33	4.31	3.92	1.77
Open outdoor	RR	Error	2.99%	3.41%	3.79%	4.43%	4.72%
		SNR(dB)	12.45	11.98	10.03	9.77	8.59
	HR	Error	4.75%	5.12%	5.47%	6.32%	6.83%
		SNR(dB)	8.32	7.25	5.29	4.03	2.71

**Table 3 sensors-22-06116-t003:** Estimation error of RR and HR of different number of subjects (unit of all values is bpm).

Vital Signs	Statistical Parameters	1 Subject	2 Subjects	3 Subjects
RR	Mean	0.0224	−0.3621	−0.5685
	SD	0.3565	1.391	2.337
	Mean − 1.96SD	−0.6764	−3.089	−5.149
	Mean + 1.96SD	0.7212	2.365	4.012
HR	Mean	−0.3074	−0.9832	−1.130
	SD	1.193	2.350	3.140
	Mean − 1.96SD	−2.646	−5.590	−7.285
	Mean + 1.96SD	2.031	3.623	5.024

**Table 4 sensors-22-06116-t004:** Errors and signal-to-noise ratios of different algorithms for estimating vital signs.

Papers	Published Year	Algorithm	RR_Error (%)	HR_Error (%)	SNR (dB)
[30]	2017	VMD Algorithm	12.76	13.58	−8.44
[17]	2019	FVPIEF	10.34	9.93	−5.24
[18]	2019	Convolutional Sparse Coding	9.81	10.77	−4.67
[10]	2020	PCA	6.69	8.52	4.19
[23]	2020	Quadrature Demodulation	5.63	5.37	3.23
This study		Proposed	5.14	4.87	6.56

## Data Availability

Data are contained within the article.

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
