# Peer review of "A Non-Contact Detection Method for Multi-Person Vital Signs Based on IR-UWB Radar"

_sensors, 2022, doi:10.3390/s22166116_

Round 1

Reviewer 1 Report

This paper introduced a non-contact multi-human vital signs detection method based on IR-UWB radar.

Before the publication, several suggestions below should be considered:

1. The author gives the full name of CIR in the abstract. However, in the main body of the manuscript, the first occurrence of CIR still needs to give the full name. Same problem as the “FT”

2. In lines 253 to 258 of the article, the author claims that the target distance detection uses RMS and EK. It is puzzling why STFT and WT are mentioned in the text, and there is an obvious discontinuity in the text.

3. The author should describe in more detail the specific parameters of the radar used and the specific signals. As far as I know, the radar is a single-input single-output (SISO) radar, which belongs to one-dimensional radar. In terms of the radar principle, one-dimensional radar can only measure distance, so why can it obtain the azimuth information of the target? At the same time, as a SISO structure, only the directivity of the transmitting antenna has been fixed, and I do not agree that beamforming can play a role in directivity differentiation at this time. As far as I know, beamforming requires 3 or more transmit antennas. From this point of view, some conclusions in this paper are debatable. We suggest that the author should give detailed information about this.

4. Some grammatical errors need to be corrected, such as in line 329.

Author Response

Dear expert: 

Thank you very much for your valuable comments, which will help us revise and improve the quality of the manuscript. Please see the attachment for the specific revisions of the manuscript. Finally, I sincerely wish you smooth work and happy life!

Reviewer 2 Report

This is a very interesting paper and the results are impressive.

You should check for minor mistakes, e.g on page 9 line 278 there is something missing.

Then it can be published.

Author Response

(The authors gave the same response as above.)

Reviewer 3 Report

In my opinion it is an interesting article but it needs to be improved. I also note that the effort in developing algorithms associated with IR-UWB could be significantly reduced with another type of radar signal and with the Doppler effect, for example.

For the present application, the biggest limitation seems to be the fact that the person is immobile, proven by experiments (fig. 3). It should clarify usage limits.

Non-contact vital sign detection has many applications, but the reference to the new coronavirus (Corona Virus Disease 2019, COVID-19), as an emerging application, seems to me to be exaggerated. This mention should be removed, at least in the abstract.

Throughout the paper there are several basic descriptions that refer to radars in general and not specifically to IR-UWB. Among others, see the example: "Since IR-UWB signals generate signal attenuation and distortion when penetrating buildings (e.g., walls), it may result in false alarms and low detection rates. In literature [7], multi-target was achieved by detecting the possibility of target presence at a location in the target's movement for localization and tracking."

References must be reviewed and properly justified or withdrawn. See the example of "IR-UWB radar signals have the main advantage of is good material penetration, which can easily penetrate for vital signs detection and target walls identification [9]." It does not clarify why the IR-UWB has this feature. In my opinion, the penetration of the signal in walls will essentially have to do with the type of material and the frequency of the wave. Furthermore, reference [9] does not address whether the IR-UWB “can easily penetrate walls for vital signs detection and target identification”.

Once again the following sentence applies to a generic radar and not specifically to an IR-UWB. "IR-UWB radar is to transmit electromagnetic waves through the radar transmitter, which reaches the surface of the human chest cavity via the propagation medium, and then reflect and scatter through the human body to form the radar echo signal. It reaches the radar receiver via the propagation medium, and then it is received by the receiver." Furthermore, writing that the transmitter radar transmits and the receiver receives does not provide useful information in the context of this paper.

"In this study, an optimized algorithm for estimating TOA with Root Mean square (RMS) and Excess Kurtosis (EK) is applied in China". China is too big and populous to be a useful reference.

Author Response

(The authors gave the same response as above.)

Reviewer 4 Report

The paper is well drafted and organized. The paper is well supported by mathematical analysis and live results. Hence I recommend to accept the work after minor revisions.

There are grammatical errors in the document and proof read the whole document to meet the standards of MDPI.

The figures 1 and 2 are not readable and I advise the authors to improve the resolution to 300 dpi.

Author Response

(The authors gave the same response as above.)

Round 2

Reviewer 1 Report

The authors have addressed all my concerns, and this work is suggested to be accepted.